# Stratospheric impact of the anomalous 2023 Canadian wildfires: the two vertical pathways of smoke

Sergey Khaykin<sup>1</sup>, Slimane Bekki<sup>1</sup>, Sophie Godin-Beekmann<sup>1</sup>, Michael D. Fromm<sup>2</sup>, Philippe Goloub<sup>3</sup>, Qiaoyun Hu<sup>3</sup>, Béatrice Josse<sup>4</sup>, Alexandra Laeng<sup>5</sup>, Mehdi Meziane<sup>4</sup>, David A. Peterson<sup>6</sup>, Sophie Pelletier<sup>4</sup>,

- 5 Valérie Thouret<sup>7</sup>
  - <sup>1</sup> LATMOS Laboratoire Atmosphère Observations Spatiales, UVSQ, CNRS, Sorbonne University, Guyancourt, France
  - <sup>2</sup> Naval Research Laboratory, Washington, DC, USA.
  - <sup>3</sup> University Lille, CNRS, LOA Laboratoire d'Optique Atmosphérique, Lille, France
  - <sup>4</sup> Centre National de Recherches Météorologiques, Université de Toulouse, Météo-France, CNRS, Toulouse, France
  - <sup>5</sup> Karlsruhe Institute of Technology, Institute of Meteorology and Climate Research (IMKASF), Karlsruhe, Germany
    - <sup>6</sup> Naval Research Laboratory, Monterey, CA, USA
    - <sup>7</sup> Laboratoire d'Aérologie, Université Toulouse III Paul Sabatier, CNRS, Toulouse, France

Correspondence to: Sergey Khaykin (sergey.khaykin@latmos.ipsl.fr)


Abstract. The climate-altering potential of wildfires through their emissions into the stratosphere has only recently been realized following the major wildfire outbreaks in Canada and Australia. The 2023 Canadian wildfire season stood out for its extended burned area and duration, by far exceeded the previous record-breaking events, including the Australian "Black Summer" in terms of the emitted power and pyroCb count with a total number of 142 Canadian pyroCb events over the season. The incessant fire activity all across Canada produced a succession of smoke injections into the lower stratosphere. Here, we use various satellite data sets, airborne and ground-based observations together with chemistry-transport model simulations to show that despite the exceptional vigor of the 2023 Canadian wildfires, the depth of their stratospheric impact was surprisingly shallow and limited to the lowermost stratosphere. Conversely, the incessant fire activity featuring a long succession of moderate-strength pyroCb events, combined with numerous episodes of synoptic-scale smoke uplift through the warm conveyor belt, led to unparalleled levels of pollution at commercial aircraft cruising altitudes throughout the season.

#### 1 Introduction









The severity of wildfires has remarkably increased in the twenty-first century in response to the regional and global warming trends (Cunningham et al., 2024; Jones et al., 2020; Virgilio et al., 2019) and there is an emerging awareness of their impact on climate and ozone layer (Bernath et al., 2022; Chang, 2021; Hirsch and Koren, 2021; Khaykin et al., 2020; Ma et al., 2024; Salawitch and McBride, 2022; Sellitto et al., 2022; Solomon et al., 2023) via injection of biomass burning emissions into the stratosphere. Intense wildfires release tremendous amounts of heat into the atmosphere, which gives rise to extreme thunderstorms termed pyrocumulonimbus (pyroCb). These storms, augmented by the energy of combustion, can generate vigorous convective updrafts injecting smoke into the stratosphere, where the residence time of aerosols is not limited by cloud scavenging and precipitation (Fromm et al., 2010; Peterson et al., 2018). A number of recent studies have put in evidence that the effects of strong pyroCb events on the global stratosphere rival those of moderate volcanic eruptions in terms of magnitude and duration (e.g. Peterson et al., 2021, 2018) whilst exceeding them in terms of radiative forcing (D'Angelo et al., 2022; Das et al., 2021; Yu et al., 2021).

In contrast to explosive volcanic eruptions injecting ash and sulphuric aerosol precursors, the pyroCb storms lift carbonaceous aerosol, including organic, brown and black carbon. Due to absorption of solar radiation by the black carbon, the smoke plumes can be propelled higher into the stratosphere by radiative heating (Allen et al., 2024, 2020; Kablick et al., 2020; Khaykin et al., 2020, 2018), which prolongs their stratospheric residence time (Yu et al., 2019).

While direct stratospheric injections by pyroCb activity have been demonstrated to be the primary source of combustion products entering the stratosphere (Allen et al., 2024; Fromm et al., 2010; Kablick et al., 2020; Katich et al., 2023; Peterson et al., 2021, 2018; Schwartz et al., 2020), other troposphere-stratosphere pathways of smoke, such as synoptic-scale uplift of warm air and radiatively driven ascent from the lower and middle troposphere, have also been invoked (Hirsch and Koren, 2021; Magaritz-Ronen and Raveh-Rubin, 2021; Ohneiser et al., 2023), however the impact of these secondary vertical transport pathways on stratospheric composition remains highly uncertain.

The 2023 wildfire season in Canada was marked by an unprecedented burned area exceeding 45 million acres, rendering it the most destructive ever recorded (Byrne et al., 2024; Jain et al., 2024). The anomalously early onset and duration of raging wildfires all across Canada, spanning early May through late September, can be paralleled with the expected rise of the fire season duration and frequency of dry years. The sustained extreme fire weather conditions were enabled in part by a warm temperature anomaly of +2.2° C over Canada as compared to the 1991-2020 average (Jain et al., 2024), which resulted from persistent blocking features that affected the synoptic weather patterns (Peterson et al., 2025).

In terms of burned area, the 2023 Canadian wildfires greatly exceeded previous record-breaking wildfire events worldwide including the Australian "Black Summer", which burned 18.3 million acres (Australian Government, 2020) and generated the Australian New Year Super Outbreak (ANYSO) of pyroCb activity. The ANYSO event caused a substantial large-scale perturbation of stratospheric aerosol and gaseous composition within a deep stratospheric layer (e.g. Khaykin et al., 2020; Peterson et al., 2021). Considering the exceptional vigor of the 2023 Canadian wildfires, one is led to expect a proportionally larger impact on the stratosphere, however, as reported by Zhang et al. (2024) the vertical extent of stratospheric perturbation was shallow. This study explores the nature, character and magnitude of the stratospheric perturbations induced by the anomalous 2023 Canadian wildfires.

## 2 Data sets and methods

# 2.1 GFAS fire radiative power

The Global Fire Assimilation System (GFAS) Fire Radiative Power (FRP) product is a satellite-derived dataset that quantifies the energy emitted by active fires globally. FRP is a key parameter for estimating fire intensity and biomass burning

emissions. GFAS assimilates FRP observations from multiple satellite missions, including the Moderate Resolution Imaging Spectroradiometer (MODIS) (Justice et al., 2002) aboard the Terra and Aqua satellites, and the Visible Infrared Imaging Radiometer Suite (VIIRS) (Polonsky et al., 2014) on the Suomi National Polar-orbiting Partnership (Suomi NPP) satellite and NOAA-20. These satellite instruments provide a comprehensive and near-real-time representation of fire activity. GFAS provides daily gridded fire emission estimates at a spatial resolution of 0.1° x 0.1° (Rémy et al., 2017). To compute the cumulative energy released by fires, the original FRP data provided in W m<sup>-2</sup> units are spatially integrated over the respective area to obtain the total energy in units of TW.

#### 2.2 PyroCb detection and inventory









All pyroCb event location and time information for 2013-2023 were obtained from a global inventory of 761 events described in (Peterson et al., 2025), which builds from an earlier version of the inventory for 2013-2021 used by Fromm et al. (2022). A brief summary of this effort is provided here. This dataset is based in part on a growing community effort to inventory all observed pyroCb activity worldwide (analyst-in-the-loop), called The Worldwide PyroCb Information Exchange (https://groups.io/g/pyrocb), which requires constant attention to fires and pyroCb activity in all regions worldwide. The inventory also leverages a previously-developed automatic pyroCb-detection algorithm that has been applied to geostationary weather satellite observations (Peterson et al., 2017b, 2017a). Data from this effort provide, to our knowledge, the only multi-year inventory of all known pyroCb activity worldwide.

All pyroCb detections require a convective cloud that remains anchored to a wildfire, as evidenced by a cluster of active fire pixels detected by satellite. Manual and automatic detections are based on the distinctive cloud microphysics of pyroCb activity when compared with traditional convection (Fromm et al., 2010; Peterson et al., 2017a; Rosenfeld et al., 2007). A pyroCb must exhibit a minimum 11 μm brightness temperature less than an approximated homogeneous liquid-water freezing threshold of -35°C to -40°C (Peterson et al., 2017a). During daytime, pyroCb detection takes advantage of unusually small particles in the pyroCb cloud tops (Chang et al., 2015; Reutter et al., 2014; Rosenfeld et al., 2007). Differences in 3.9 μm and 11.0 μm brightness temperature become unusually large (near and greater than 50 K) in the presence of such smaller particles (Fromm et al., 2010; Peterson et al., 2017a), allowing pyroCbs to be separated from other deep convection.

Other criteria for pyroCb detection include an optically thick (opaque) cloud core (Peterson et al., 2017a) and reduced visible reflectance when compared with traditional thunderstorm cloud tops (Rosenfeld et al., 2007). Weather radar echo-tops are employed to characterize pyroCb injection altitude when and where data are available (e.g., (Fromm et al., 2021; Peterson et al., 2021)). The confidence of each pyroCb detection is augmented with ultra-violet absorbing aerosol index (UV AAI) (Guan et al., 2010; Torres et al., 1998), lidar backscatter profiles, and backward trajectory calculations. All entries in the inventory are listed at the pyroCb 'event' level, defined as an individual pyroCb pulse or chain of several pulses (and resulting smoke injections) linked to a specific fire or segment of a large fire front (Peterson et al., 2021).

## 2.3 TROPOMI

The TROPOspheric Monitoring Instrument (TROPOMI), aboard the Sentinel 5 Precursor mission, is a nadir-viewing shortwave spectrometer developed by the Netherlands Space Office and the European Space Agency. Among its measurements, the Absorbing Aerosol Index (AAI) is a key parameter derived from ultraviolet (UV) spectral bands (340–380 nm) (Veefkind et al., 2012). AAI is calculated using the spectral contrast between a pair of UV wavelengths, based on the ratio of the observed top-of-atmosphere reflectance and a pre-calculated theoretical reflectance for a Rayleigh-scattering-only atmosphere (Torres et al., 1998). Positive AAI values indicate the presence of UV-absorbing aerosols, such as dust and smoke. AI is influenced by aerosol properties, including optical thickness, single scattering albedo, as well as the aerosol layer height. TROPOMI provides global coverage and a high spatial resolution of 7×3.5 km² at nadir.

#### 2.4 OMPS Nadir Mapper









The Ozone Mapping and Profiler Suite Nadir Mapper (OMPS-NM) is a spectrometer designed to provide global observations of atmospheric ozone and other trace gases. Aboard the Suomi National Polar-orbiting Partnership (SNPP) satellite, operational since 2012, OMPS-NM measures backscattered solar radiation in the ultraviolet (UV spectral region (300–380 nm) (Flynn et al., 2014). This spectral range enables retrievals of total column ozone and the absorbing aerosol index (AAI), a key parameter for detecting UV-absorbing aerosols such as smoke and dust (Torres et al., 1998). OMPS-NM offers a spatial resolution of approximately 50 km×50 km at nadir, allowing detailed mapping of ozone distributions and aerosol features on a global scale (Jaross, 2014). Its cross-track scanning capability ensures near-global coverage in a single day, making it a valuable tool for monitoring atmospheric composition and detecting events like volcanic eruptions and large-scale biomass burning.

## 2.5 OMPS Limb Profiler

The Ozone Mapping and Profiler Suite Limb Profiler (OMPS-LP) on the Suomi National Polar-orbiting Partnership (Suomi-NPP) satellite, operational since April 2012, observes limb scattered sunlight in the 290–1000 nm spectral range (Jaross, 2014). The sensor employs three vertical slits separated horizontally to provide near-global coverage in 3 - 4 days and >7000 profiles a day. The instrument achieves a vertical resolution of approximately 1.5 km, whereas the accuracy of extinction profiles is 10–20% depending on the altitude. Here we use OMPS-LP NASA V2.1 cloud-unfiltered aerosol extinction profiles at 869 nm (Taha et al., 2021) and layer cloud/aerosol flagging data for analysis of extinction ratio profiles, spatiotemporal tracking of aerosol plumes. The 869 nm channel is chosen because it showed the best agreement with SAGE III data (Taha et al., 2021). Extinction ratio is computed as the ratio between aerosol and molecular extinction.

# 2.6 Stratospheric Aerosol Layer Detection (SALD)

OMPS-LP V2.0 data include information on the cloud height and type derived from the ratio of measured to calculated radiances ratio. Cloud type classifies the identified cloud as tropospheric cloud, enhanced aerosol or polar stratospheric cloud (PSC). The enhanced aerosol definition requires the cloud altitude to be at least 1.5 km above the tropopause (Taha et al., 2021). We combine these data with OMPS-LP Stratospheric Aerosol Optical Depth (SAOD) and Extinction Ratio (ER) to introduce the Stratospheric Aerosol Layer Detection (SALD), which is used to track the stratospheric aerosol plumes in time and space. SALD is defined as an event flagged as enhanced aerosol. In order to minimize false detections, we apply additional filtering using the empirically determined minimum thresholds of 0.01 for SAOD and 8 for ER, roughly corresponding to seven standard deviations of the zonal-mean values in the non-perturbed conditions. The SALDs contain the information on the plume top altitude (derived directly from the original cloud height field and plume peak altitude defined as the altitude of the maximum ER). Note that while the plume top altitude of SALD is always above the local tropopause, the plume peak altitude may be below the tropopause.

## 2.7 Estimation of mass of injected smoke aerosols

Mass of smoke aerosols injected into the stratosphere was estimated using OMPS-LP extinction profiling data and the absolute mass difference method (Khaykin et al., 2020) with the assumed particle mass extinction coefficient of  $4.5 \,\mathrm{m^2\,g^{-1}}$  (Peterson et al., 2018). The daily mass of aerosols is computed by integrating the aerosol extinction in horizontal and vertical dimensions within the latitude band affected by wildfires ( $40^{\circ} \,\mathrm{N-82^{\circ}\,N}$ ) and within the altitude layer where smoke aerosols were detected i.e., between the tropopause and  $16 \,\mathrm{km}$  (Fig. S10). After converting the integrated extinction to mass, the resulting daily time series of aerosol mass are smoothed using 7-day boxcar. To compute the injected mass corresponding to

specific event, the aerosol mass on the day before the event is compared with the local maximum of mass following the event (Fig. S10). This difference is considered to be due to the smoke uplifted into the stratosphere. The error bar on the aerosol mass takes also into account the uncertainty on the particle mass extinction coefficient (1.5 m<sup>2</sup> g<sup>-1</sup>). The main limitation of this method is linked with the variability of stratospheric aerosol load modulated by volcanic eruptions and meridional transport of aerosols. In Summer 2023, the global stratosphere was affected by the Hunga eruption in January 2022 (Tonga) (Khaykin et al., 2024). A gradual removal of volcanic aerosols from the extratropical stratosphere by sedimentation and horizontal transport resulted in a progressive decay of its SAOD throughout the wildfire season (Fig. S10), which reduced the difference between the pre-event and post-event stratospheric aerosol mass. The obtained injected masses represent thus a lower-bound estimate.

## 2.8 SAGE III /ISS stratospheric aerosol extinction








The Stratospheric Aerosol and Gas Experiment (SAGE) III provides stratospheric aerosol extinction coefficient profiles using solar occultation observations from the International Space Station (ISS) (Cisewski et al., 2014). These measurements, available since February 2017, are provided for nine wavelength bands from 385 to 1550 nm and have a vertical resolution of ~0.7 km and are characterized by high precision (<5%). We use version V5.3 of SAGE III solar occultation aerosol extinction data at 869 nm. Only the data above the local thermal tropopause (derived from MERRA-2 reanalysis) are used for plotting.

## 2.9 MOCAGE Chemistry-transport model simulation

MOCAGE (Modèle de Chimie Atmosphérique de Grande Echelle) is the chemistry-transport model developed by Météo-France (Cussac et al., 2020; Guth et al., 2016; Josse et al., 2004). It is used for a large number of research studies into atmospheric composition (gases and aerosols) on global and regional scales. It is also used routinely on a daily basis, both to forecast global composition and over an extended Europe at higher resolution. The model describes the gaseous chemical composition of the troposphere and stratosphere by merging the RACM (Stockwell et al., 1997) and REPROBUS (Lefèvre et al., 1994) schemes, including 110 species and 394 reactions. The primary aerosols taken into account are desert dust, sea salts, soot and organic carbons. Secondary inorganic aerosols follow the representation of Guth et al. (2018; 2016) and secondary organic aerosols a simplified representation (Descheemaecker et al., 2019). Each of the aerosols is represented on 6 bins.

The 60 vertical levels follow a sigma-pressure coordinate, and extend from the ground up to 0.1 hPa, or about 60km. Furthermore, in this study, the horizontal resolution of the model is set at 0.5 degrees longitude x 0.5 degrees latitude on the globe. As MOCAGE is a CTM, meteorological variables are provided as inputs. In this study, the operational numerical weather prediction model ARPEGE (Courtier et al., 1991) is used. Large-scale transport is based on a semi-Lagrangian scheme (Williamson and Rasch, 1989), and turbulent convection and diffusion are parameterised according to Bechtold et al. (2001) and (Louis, 1979) respectively. It is important for this study to emphasise that no parameterisation of pyroconvection is implemented in the model.

Emissions are for the most part derived from static inventories, in this case the CAMS inventory (CAMS, 2020). However, desert dust and sea salt emissions are calculated dynamically, in particular as a function of wind. Carbonaceous aerosols come from two sources: anthropogenic emissions, which are listed in the inventories, and emissions from biomass fires. For the latter, we use hourly data provided by GFAS. The information used is the quantities injected. However, in this study, the use of plume height as provided by GFAS was not activated: all biomass burning emissions were injected from the surface to an altitude of 2km. This avoids any suspicion of pseudo-parametrisation of pyroconvection.

Finally, MOCAGE has an observation assimilation module. Here, we use the model's ability to assimilate Aerosol Optical Depth (AOD) from MODIS (Moderate-Resolution Imaging Spectroradiometer), as described by El Amraoui et al. (2022); Sič et al. (2015). It is important to note that these AOD observations correspond to vertically integrated content. Therefore, assimilation will be able to modify the total amount of aerosols represented by the model, but in no case the vertical distribution.

## 2.10 IAGOS airborne observations of CO and O<sub>3</sub>

IAGOS (In-service Aircraft for a Global Observing System; http://www.iagos.org) is a European Research Infrastructure for global observations of atmospheric composition from commercial aircraft. The objective is to provide essential data on climate change and air quality at a global scale (Petzold et al., 2017; Thouret et al., 2022). Indeed, the use of commercial aircraft (10 in operations in 2024) allows the collection of highly relevant observations on a scale and in numbers impossible to achieve using research aircraft, and where other measurement methods (e.g., satellites) have technical limitations. IAGOS provides a database for users in science and policy, including near real time data provision for weather prediction and air quality forecasting. IAGOS data are being used by researchers world-wide for process studies, trend analysis, validation of climate and air quality models, and the validation of space borne data retrievals. Among the various atmospheric compounds recorded by IAGOS equipped aircraft, the one used in this analysis is the CO dataset. CO measurements are performed by an Infra-Red correlation automatic analyser as described in detail by Nédélec et al., (2015). The assessment of the quality and long-term stability of this data set is further described by Blot et al. (2021).

## 2.11 LILAS lidar








LILAS is a multi-wavelength lidar system operated at ATOLL observatory (50.6°N, 3.1°E, 60 m) in northern France. LILAS utilizes an Nd:YAG laser emitting at three wavelengths: 355, 532 and 1064 nm, with a repetition rate of 20 Hz. The backscattered light is collected with a 40 cm telescope. The optical reception module includes detection channels for the three elastic scattering wavelengths and three Raman scattering wavelengths – 387 nm (vibrational Raman of N2), 408 nm (vibrotational Raman H<sub>2</sub>O vapor) and 530 nm (rotational Raman of N<sub>2</sub> and O<sub>2</sub>). In addition, a broadband fluorescence channel centered at 466 nm has been integrated to LILAS, providing high sensitivity to bioaerosols. The lidar signals are recorded with Licel transient recorders with a range resolution of 7.5 m and a time resolution of 1 minute. The configuration of LILAS allows the acquisition of vertical profiles of the extinction and backscatter coefficients, linear particle depolarization ratios, water vapor mixing ratio and relative humidity, fluorescence backscattering coefficient and fluorescence capacity. The operation and calibration of LILAS are conducted following the guidance and standards of EARLiNET (European Aerosol Research Lidar Network), one of the remote sensing component of the ACTRIS (Aerosol Cloud Trace gas Research Infra Structure) infrastructure. Further details in the LILAS instrument are provided in (Hu et al., 2019) and references therein.

## 2.12 OHP LTA lidar

The Observatoire de Haute-Provence (OHP) located in southern France (43.9° N, 5.7° E, 670 m) is equipped with several lidar systems for atmospheric sounding at a wide range of altitudes. The aerosol measurements are provided by LTA (Lidar Température, Aérosol)instrument operating at 532 nm since 1991 on a regular basis with a mean measurement rate of 10-12 acquisition nights per month. For retrieving vertical profiles of stratospheric aerosol, we apply Fernald-Klett inversion method, which provides backscatter and extinction coefficients. The scattering ratio is then computed as a ratio of total (molecular plus aerosol) to molecular backscattering, where the latter is derived from ECMWF meteorological analysis. The resulting vertical

profiles of aerosol parameters are reported at 150 m vertical resolution. A more detailed description of the instruments, aerosol retrieval and error budget are provided in Khaykin et al. (2017) and references therein.

## 2.13 Integration of data sources

In this study, pyroCb detections from the global inventory were combined with satellite observations from OMPS-NM, TROPOMI, and OMPS-LP to track smoke injection and transport. The stratospheric extinction profiles from OMPS-LP and SAGE III/ISS were used to constrain the large-scale aerosol perturbation. Ground-based lidar (LILAS, OHP) and radiosonde profiles provided high-resolution vertical structure, while IAGOS in situ aircraft data supplied CO and O3 measurements for characterization of plume chemical composition. These observational datasets were combined with fire emissions from GFAS and compared against MOCAGE chemistry-transport simulations (with MODIS AOD assimilation) to evaluate injection heights, aerosol loading, and plume dispersion. This integrated workflow provides a consistent observational–model framework for analyzing the evolution of wildfire smoke in the lower stratosphere.

## 3 Results







## 3.1 The anomalous 2023 Canadian wildfire season

The 2023 Canadian wildfire season can be characterized by incessant flaming fires from early May through late September. Figure 1A shows the cumulative energy generated by the wildfires as derived from the fire radiative power (W m<sup>-2</sup>) provided by the Global Fire Assimilation System (GFAS). The cumulative energy was steadily increasing throughout the season and surpassed the Australian "Black Summer" benchmark (135 TW h), as well as all previous North American records already by early July. By the end of the wildfire season, the cumulative fire energy has reached 200 TW h (0.7 EJ), which is more than a factor of two larger than the annual energy production by Canadian nuclear plants (Statistics Canada, 2024).

**Figure 1.** General metrics of the 2023 Canadian wildfires in perspective. A) Cumulative energy (in TWh) released by wildfires Canada from May through October for different years since 2003 from GFAS data. (B) Cumulative number of PyroCb events in Canada since 2013. C) Seasonal variation (May through October) of the maximum Absorbing Aerosol Index (AAI<sub>max</sub>) over Canada from OMPS-NM observations since 2012. Black circles mark the events with AAI<sub>max</sub>>15 associated with stratospheric injection of smoke.

In terms of pyroCb activity, the 2023 Canadian wildfires have surpassed all previous benchmarks worldwide with a total number of 142 Canadian pyroCb events over the season. The average frequency of pyroCbs across Canada amounted to 1 d<sup>-1</sup> during May-June, increasing to more than 2 d<sup>-1</sup> in July and decreasing to only a few events in August-September (Fig. 1B).

A convenient first-order proxy for the amount of smoke emitted into the upper troposphere and lower stratosphere (UTLS) is the UV Absorbing Aerosol Index (AAI, dimensionless) measured by a number of satellite nadir sensors. AAI is sensitive to both the amount and the altitude of absorbing particles, such as brown and black carbon (Torres et al., 2007), and the values exceeding 15 are conventionally associated with injection of smoke into the stratosphere (Fromm et al., 2010, 2008, 2021; Peterson et al., 2021, 2018).

To put the 2023 wildfires in perspective, Figure 1C shows the seasonal variation of the maximum AAI (AAI<sub>max</sub>) over North America since 2012 from Ozone Mapping and Profiler Suite Nadir Mapper (OMPS-NM) (Flynn et al., 2014) observations. The black circles indicate the events with AAI<sub>max</sub> exceeding 15, which are expected to represent stratospheric injections (Peterson et al., 2018). These include the well documented Pacific Northwest Event (PNE) in August 2017 (Fromm et al., 2021; Khaykin et al., 2018; Peterson et al., 2018), the Californian Creek fire in September 2020 (Hu et al., 2022; Lareau et al., 2022) as well as other events. The 2023 Canadian wildfires produced five cases with AAI<sub>max</sub>>15. Surprisingly, four of them occurred during August-September, when the pyroCb frequency was relatively low (cf. Fig. 1B).

## 3.2 Succession of wildfire and pyroCb events

In order to describe the succession of wildfire events and characterize their impact on the stratosphere, we combine the pyroCb inventory derived from geostationary imaging (Peterson et al., 2025), AAI measurements by OMPS-NM and aerosol extinction profiling by OMPS-LP (Limb Profiler) (Jaross, 2014). The NASA OMPS-LP retrieval algorithm (Taha et al., 2021) provides the top height of the detected cloud/aerosol layers, which are classified as stratospheric aerosol if the layer's top exceeds the tropopause height by 1.5 km. We apply additional filtering to these data to minimize false detections (Sect. 2.6) and refer to the resulting product as Stratospheric Aerosol Layer Detections (SALD). Considering the westerly zonal flow in the summertime midlatitude stratosphere, SALD data enable tracking of the stratospheric plumes from a given high-AAI event in the time-longitude dimension and evaluate the stratospheric plume lifetime.

Seven events during May-September 2023 with a measurable stratospheric impact have been identified, of which six began in Canada and one in eastern Siberia, as summarized in Table 1. Figure 2A displays the zonal evolution of AAI<sub>max</sub> within the 40° N - 90° N latitude band with the AAI<sub>max</sub>>15 cases encircled. Individual pyroCb events are marked by small triangles, whereas the pyroCb cluster events (involving 3 or more individual pyroCbs occurring within a 3° x 3° deg. domain and 24 hours) are displayed as large triangles. PyroCb clusters were previously associated with the largest stratospheric injections (Peterson et al., 2021).

Canada's 2023 pyroCb record begins with 3 events in Alberta on 4 May and a cluster of 4 pyroCbs on 5 May, producing an AAI<sub>max</sub> value of 18.1 on 6 May. The enhanced AAI values, propagating eastward as two separate plumes, can be tracked until 20 May (Fig. 2A). The corresponding stratospheric aerosol plume, represented by SALD (altitude color-coded circles) in Fig. 2B, circumnavigated the globe more than twice at a persistent altitude range between 11 and 13 km, which can be followed until early June.







| Event | Date   | Source          | Uplift               | AAImax | SALD     | SALD      | Injected mass |
|-------|--------|-----------------|----------------------|--------|----------|-----------|---------------|
| #     | UTC    | Location        | mechanism            |        | altitude | lifetime  | (Gg)          |
|       |        |                 |                      |        | (km)     | (days)    |               |
| 1     | 5 May  | Alberta         | PyroCb cluster       | 18.1   | 9 - 13   | >21       |               |
| 2     | 30 Jun | Eastern Siberia | PyroCb twin          | 11.5   | 12 - 15  | 17        |               |
| 3     | 14 Aug | NWT, Canada     | WCB                  | 19.8   | 9 - 12   | 28        | 16±5          |
|       |        |                 |                      |        |          |           |               |
| 4     | 26 Aug | BC, Canada      | WCB                  | 13.6   | 9 - 10   | Uncertain |               |
| 5     | 2 Sep  | BC, Canada      | WCB                  | 19.2   | 9 - 11   | >21       | 7±2           |
| 6     | 15 Sep | BC/Alberta      | PyroCb + WCB         | 20.2   | 9 - 12   | 31        | 17±6*         |
| 7     | 22 Sep | Alberta/BC      | PyroCb cluster + WCB | 18.6   | 9 - 12   | 13        |               |
|       |        |                 | ,, CD                |        |          |           |               |

**Table 1.** List of 2023 wildfire events producing smoke plumes at and above the tropopause, including event number; date; source location (pyroCb or AAI<sub>max</sub>>10); uplift mechanism (pyroCb or WCB – Warm Conveyor Belt); AAI<sub>max</sub> value; altitude range of stratospheric aerosol layer detections (SALD) by OMPS-LP; SALD temporal extent derived from Hovmoller analysis in Fig. 2; estimated aerosol mass uplifted into the stratosphere (Gg). The injected masses were estimated using OMPS-LP extinction data and the absolute mass difference method (Khaykin et al., 2020). Estimates for the events #1, #2 and #4 could not be obtained due to limitations of the method and small magnitude of stratospheric impact of these events. The injected mass for the event #6 should be considered as the sum of masses injected by #6 and #7 events that occurred close in time. See Sect. 2.7 for details on the injected mass estimation.

The subsequent pyroCb clusters occurring during the May-July period did not produce AAI<sub>max</sub> greater than 15 nor the continuous stratospheric plumes. The presence of stratospheric aerosol layers between 12-16 km altitude during the first half of July can be sourced to a twin pyroCb event in eastern Siberia (Magadan region) on 30 June, which despite relatively low AAI<sub>max</sub> value (11.5) produced a continuous stratospheric plume that was detected by lidars over France on 14 July at 13 -15 km altitude (Fig. S1). Further support for the attribution of stratospheric plumes to specific wildfires is available as a sequence of daily AAI maps with SALD and pyroCb locations in Supplementary Animation 1.

The second AAI<sub>max</sub>>15 event occurred on 14 August and produced an intense stratospheric plume at altitudes between 10 - 13 km. Surprisingly, this event was not associated with pyroCb activity, as can be inferred from Figs. 2A and 2C. The absence of pyroCb was equally the case for the successive AAI<sub>max</sub>>15 event on 2 September that produced persistent stratospheric aerosol plume. The later two AAI<sub>max</sub>>15 events were linked respectively to an individual pyroCb event on 15 September and to a pyroCb cluster on 22 September. Both events occurred near the border between Alberta and British Columbia.

The widespread stratospheric impact of the August-September events is evident in Fig. 2B and 2D. The succession of wildfires producing stratospheric plumes resulted in nearly complete zonal spread of smoke throughout the 40° N - 90° N latitude band in late September - early October. The significant stratospheric impact of the wildfire events that did not involve pyroCb injections led us to explore the non-pyroCb mechanisms of smoke uplift.

**Figure 2.** Spatiotemporal evolution of smoke plumes during the 2023 wildfire season. (A) Longitude-time variation of AAI<sub>max</sub> within 40° N - 90° N. Black circles mark the events with AAI<sub>max</sub>>15. Small open and large filled triangles indicate respectively the individual and cluster PyroCb events. (B) Longitude-time variation of OMPS-LP SALD (Stratospheric Aerosol Layer Detection) within 40° N - 90° N displayed as circles color-coded by the top altitude of aerosol layer. The underlying image shows AAI<sub>max</sub> (same as B). (C and D) As in (A) and (B) but in latitude-time space with full zonal coverage.

## 3.3 Pathways for vertical smoke transport

365

370

395

Self-lofting of wildfire smoke in the stratosphere has been reported by a number of studies focusing on 2009 Australian "Black Saturday" (Allen et al., 2024), 2017 Canadian PNE (Khaykin et al., 2018; Lestrelin et al., 2021; Yu et al., 2019) and the 2019/2020 Australian "Black Summer" ANYSO events (Kablick et al., 2020; Khaykin et al., 2020). In each case, the self-lofting of the biomass burning plume was associated with a persistent stratospheric anticyclone (SCV - Smoke-Charged Vortex, or SWIRL - Smoke with Induced Rotation and Lofting) that provided dynamical confinement to the plume thereby maintaining light-absorbing aerosols at high concentration and high degree of their internal heating. A few studies have invoked radiatively-driven ascent of smoke from the lower/middle troposphere to the stratosphere (Laat et al., 2012; Ohneiser et al., 2023), however their analysis did not rule out direct pyroCb injections as the source of observed stratospheric smoke.

**Figure 3.** WCB-driven smoke uplift episode (event #3) illustrated as sequential geographical maps for the 13 - 17 August 2023 period. (A - E) Color shading show MOCAGE-simulated altitude of maximum concentration of wildfire aerosols (km); black contours show ERA5 geopotential height at 500 hPa (labels in dam); pink contour with grey shading indicates areas with downward ERA5 500 hPa vertical velocity; open circles mark OMPS-LP ground track locations; color-filled circles indicate OMPS-LP SALD locations (same color map as for MOCAGE altitude). SALD altitude corresponds to the peak of the observed extinction ratio profile. All maps are provided for 18 UT, which roughly corresponds to the time of OMPS-LP measurement within the given region. (F) TROPOMI aerosol index on 17 August 2023 with ERA5 500 hPa geopotential and OMPS-LP SALDs.

Another mechanism of air uplift from the lower troposphere is the warm conveyor belt (WCB), a synoptic process capable of lofting air into the upper troposphere within the warm sector of a mid-latitude cyclone on a scale of a few days (Eckhardt et al., 2004). This WCB pathway for stratospheric smoke injection is explored using MOCAGE chemistry-transport model is constrained by daily GFAS emissions with injection height set to 2 km (Sect. 2.9). The simulation does not assimilate vertically-resolved observations, nor the pyroCb information.

The 14 August WCB event that produced a persistent stratospheric plume (AAI<sub>max</sub>>15) in the absence of a pyroCb source is examined in Fig. 3 for the period of 13-17 August 2023. True color satellite imagery corresponding with time periods in Fig. 3 is provided in Figure S2. These figures reveal that a large mid-latitude cyclone was located over northern Canada for the duration of this period. Its eastward progress was restricted by the development of a blocking pattern in the middle and upper troposphere (omega block) that became especially evident by 16 August. The cyclone was not tilted with height (i.e., vertically-stacked), with an occluded area of low pressure at the surface directly underneath the upper-level low (Fig. S3).

On 13 August 2023, many wildfires were burning intensely in northwestern Canada as can be inferred from a large cluster of GFAS thermal anomalies (red circles in Fig. 3A). The smoke released by these fires (blue shading) was transported to the east within the developing warm sector of the surface low pressure (Fig. S3). By 14 August (Fig. 3B), the surface low began to occlude, while the smoke plume entered a region of strong upward motion within the WCB near Hudson Bay (pink contours and grey shading). MOCAGE simulations show that smoke reached altitudes of 8-9 km on 15 August (Fig. 3C, green shading) as the smoke exited the WCB over northern Canada, corresponding with smoke visible above the cloud tops in Fig. S2. This region of upper-level diffluent winds (geostrophic flow) caused a portion of the lofted smoke plume to be transported to the northwest around the upper level low, while another portion of the plume travelled to the northeast over the high-pressure ridge of the omega block pattern at altitudes of 8-10 km during 16-17 August (Fig. 3D,E).

The highest-altitude plumes above the regional-average dynamical tropopause (10.3 km) are coincident with OMPS-LP SALDs, shown as altitude-coded circles in Fig. 3F. By 16-17 August (Fig. 3E,F), SALDs resulting from this uplift event were widespread across the Canadian Arctic and North Atlantic, well downwind of the stationary cyclone. Backward trajectories initialized from a cluster of SALDs on 16 August west of Greenland generally intersect the boundary layer above the wildfires observed on 13-14 August (Fig. S4), further supporting WCB uplift. A qualitative comparison of the simulated and observed smoke plume on 17 August, i.e. 3 days after the AAI<sub>max</sub>>15 event, is provided in Figs. 3E and 3F. The model successfully reproduces the complex shape of the plume, characteristic of WCB pattern after its frontal occlusion (Schultz and Vaughan, 2011).

Figure 4 provides a height-resolved time series of the maximum wildfire aerosol concentration from MOCAGE for the primary WCB-affected region, extending from Alaska to Europe. It reveals five successive episodes of smoke injection into the UTLS during August and September that involved the WCB mechanism (see Table 1). The 14 August event analysed in Fig. 3 stands out as the largest uplift of smoke from 4 km to 11 km, extending above the dynamical tropopause (i.e., 3.5 PVU). Stratospheric injection is confirmed by satellite observations (OMPS-LP SALDs, red circles in Fig. 4). This event and the next two WCB uplift episodes occurred in the absence of pyroCb activity in Canada, which diminished substantially during the 9 August - 13 September period.

**Figure 4.** Succession of WCB-driven uplift episodes during August-September 2023 from MOCAGE simulation. Color map shows height-resolved time series of maximum wildfire aerosol concentration within the region of WCB uplifting (40° N - 85° N, 140° W - 60°E). Red-filled circles indicate OMPS-LP SALDs (stratospheric aerosol layer detection) north of 40° N for any longitude. Solid red arrows with corresponding numbers indicate the dates of events listed in Table 1. Sloped dashed arrows illustrate the timescale of WCB uplift episodes. Black and grey curves indicate the altitude of dynamical tropopause defined as 3.5 and 2 PVU levels.

PyroCb activity resumed on 15 September, with two pyroCbs in British Columbia. However, the role of this pyroCb events in cross-tropopause smoke transport is unclear. The first post-event stratospheric detections of smoke associated with the AAI plume, emerged only on 18 September, which is three days after pyroCb cessation (Supplementary animation 1). MOCAGE simulations show gradual uplift of the smoke plume over several days preceding new detections of stratospheric smoke up to 12.5 km. The sloped dashed arrows in Fig. 4 illustrate the timescale of WCB uplift episodes.

The last AAI<sub>max</sub>>15 event in the 2023 season was linked to a pyroCb cluster event on 22 September with the maximum cloud top height reaching 12.5 km, as inferred from satellite-derived brightness temperature and a nearby radiosonde (see Sect. 2.2). A careful examination of the daily AAI and SALD maps (Supplementary animation 1) suggests that the bulk of the high AAI plume remained below the tropopause and exhibited indications of WCB-driven uplift limited to the upper troposphere, which is corroborated by MOCAGE simulation.

A basic meteorological analysis of these additional smoke uplift events involving the WCB or combined WCB and pyroCb pathways is provided in Supplementary animation 2 and Figures S5 to S8. Each of these cases generally corresponds with meteorology that is similar to the 14 August WCB event. However, differences do exist, such as the progression of the synoptic weather features and magnitude of the injected smoke plumes. The strength and position of the upper-level (500 hPa) disturbance, surface low pressure, and WCB vertical motion also vary between these cases, all of which can have an impact on potential smoke uplift and transport. Future work is required to examine the remainder of these smoke uplift events in more detail, including isolating the relative impact of WCB uplift and direct pyroCb injection for the cases on 15 and 22 September.

## 3.4 Evolution of plumes injected by pyroCb and WCB







The timescale of pyroCb stratospheric injection is typically a few hours, which owes to the fast convective uplift (Peterson et al., 2022; Rodriguez et al., 2020). An intense cloud of smoke and ice at stratospheric altitudes can be observed already on the next day after the event (Khaykin et al., 2020; Peterson et al., 2021, 2018). In contrast, a synoptic-scale uplift through the WCB mechanism requires about two days to climb to the tropopause (Eckhardt et al., 2004). As was inferred from MOCAGE simulation (Fig. 4), the WCB uplift rate from the middle troposphere to the lowermost stratosphere lies between 0.5 - 1.2 km

day<sup>-1</sup>. This is faster than radiatively-driven uplift of intense smoke plumes in the stratosphere, which barely reaches 0.5 km day<sup>-1</sup> (Khaykin et al., 2020; Lestrelin et al., 2021; Ohneiser et al., 2023). It should be noted that while the simulation does not account for the solar heating of absorbing aerosols, the simulated timescale of cross-tropopause uplift is confirmed by OMPS-LP observations, reporting the occurrence of aerosol layers above the tropopause in time with the simulated uplift across the tropopause.




The question that arises is whether the mechanism and timescale of the smoke uplift can affect the habits of the stratospheric plumes. Figure 5 compares the vertical profiles of OMPS-LP Extinction Ratio (ER, ratio between aerosol and molecular extinction) within the pyroCb- and WCB-generated aerosol plumes over the course of two weeks following the respective event. The pyroCb plumes from a cluster pyroCb event on 5 May in Alberta (Fig. 5A) as well as from a twin pyroCb event on 30 June in Eastern Siberia (Fig. 5B) can be characterized by a strong variability of the peak ER value and its potential temperature level over time. In contrast, the WCB plumes from two uplift episodes in August and September (Fig. 5C,D) do not show significant variability either in the peak ER values or in their vertical structure.

Figure 5. Vertical profiles of OMPS-LP extinction ratio at 869 nm observed after PyroCb event on (A) 5 May (event #1) and (B) 30 June (event #2) as well as after WCB uplift episodes starting on (C) 14 August (event #3) and (D) 1 September (event #5). The color of profiles indicates the age of the plume. The tropospheric parts of profiles are grey-colored. Selection of profiles is done on the basis of the SALD Hovmoller analysis (Fig. 2).

Unlike the highly variable pyroCb-generated smoke layers, the WCB plumes in the UTLS appear homogeneous in time and space and feature relatively low aerosol concentrations with the maximum ER around 12 (Fig 5 C,D). This may be

attributed to the longer timescale of smoke uplift to the tropopause through WCB process (2 - 4 days), in which the aerosols enter the stratosphere already well mixed and diluted. The low concentration of aerosols in the WCB plumes limits the degree of internal heating and thereby does not enable diabatic self-lofting in the stratosphere. Indeed, radiative transfer simulations by Ohneiser et al. (2023) showed that the lofting rate strongly depends on the smoke plume's AOD.

The differences between the pyroCb and WCB plumes can be explained using the following considerations. First, the pyroCb plumes are produced by a localized convective injection and the core of the stratospheric cloud of smoke and ice tends to remain compact (Allen et al., 2024, 2020; Kablick et al., 2020; Khaykin et al., 2020), which leads to high aerosol concentration in the young plume. The compact size may also lead to satellite undersampling of the core plume, which could partly explain the strong variability of the observed peak ER values. Apart from that, the intense stratospheric plumes produced by pyroCb injections are typically subject to diabatic self-lofting due to absorption of solar radiation by black carbon (e.g. Yu et al. (2019)). Such self-lofting is reflected in the temporal evolution of the Siberian pyroCb plume in terms of its potential temperature level (Fig. 5B). Interestingly, the plume produced by the cluster pyroCb event in Alberta (Fig. 5A) does not show diabatic self-lofting, and appears to be settling downward. A possible explanation for such behavior is the relatively low concentrations of absorbing aerosols in the plume (peak ER of 21 for Alberta plume compared to 41 for the Siberian plume) and hence the lack of internal heating. For comparison, the peak ER values of the young PNE plumes reached 74.

# 3.5 Airborne and ground-based observations of Canadian smoke









The lack of self-lofting of Canadian wildfire plumes has limited their vertical extent to the so-called Extratropical Tropopause Layer (ExTL) (Gettelman et al., 2011) and more specifically to commercial aircraft cruising altitudes (~10 - 12 km). Here, we exploit in situ airborne measurements provided by the In-service Aircraft for a Global Observing System (IAGOS) (Thouret et al., 2022). The IAGOS fleet of 10 commercial aircrafts carries various in situ sensor packages onboard, including carbon monoxide and ozone sensors.

During the active wildfire season in 2023, May through September, the IAGOS flights covered a total travel distance of 4.3 million km at cruising altitudes within the outflow region of Canadian wildfires (40°N - 90°N, 130°W - 30°E), out of which 8244 km (0.19 %), that is ~34 hours of flight time, was spent in conditions with CO concentration exceeding +3 sigma limit (195 ppbv, computed from the ensemble of cruise data). The percentage of transatlantic IAGOS flights affected by enhanced CO concentration amounts to 0.19%, which is a factor of 3 higher than the 21-yr average percentage of 0.06% (Fig. S9).

Figure 6 shows two examples of transatlantic flights sampling intense smoke plumes from high-resolution Sentinel 5P TROPOMI AAI observations. The first case of 11 May 2023 corresponds to a flight from Montreal that crossed a 6-day old plume originating from the cluster pyroCb event in Alberta on 5 May (#1 in Table 1). The flight track across the high-AAI plume over Nova Scotia is shown in Fig. 6A, whereas the time series of GPS altitude, CO and O3 mixing ratio along the A-B flight segment are shown in Fig. 6B. Shortly after reaching the cruise altitudes and crossing the dynamical tropopause (2 PVU), the aircraft was exposed to high CO mixing ratios reaching 601±39 ppbv. The CO enhancements are correlated with substantial dips in ozone mixing ratio, depleted by a factor of 4 with respect to the extra-plume environment. The ozone depletion within the smoke plumes has been reported by Bernath et al. (2022); Solomon et al. (2023); Ohneiser et al. (2021) and can be associated with transport and/or chemical processes.

Another CO enhancement up to 300 ppbv, also featuring a dip in ozone, was detected 1.5 hours later (just before 03:30 UTC) corresponding with a long filament stemming from the southern flank of the core plume (Fig. 6A). A similar filament can also be observed near the plume's northern flank. The compact shape of the plume and the counterclockwise filamentation is indicative of the anticyclonic rotation of the plume (Khaykin et al., 2022), and suggests a SCV-like self-confined structure, usually associated with massive pyroCb injections (Allen et al., 2020; Khaykin et al., 2020).

The second case of 29 August (Fig. 6C, D) corresponds to a flight from Frankfurt sampling a dense plume from a WCB event in British Columbia on 26 August (#4 in Table 1). The plume was subject to a very rapid transatlantic transport and approached Europe in under three days. Figure 6D shows CO enhancement reaching 736±42 ppbv, that is a factor of 7 higher than the background level. This is the second highest value observed during the 2023 wildfire season; the 2023 maximum of 793±45 ppbv was measured inside the WCB/pyroCb plume from the 15 September (#6) event. The dense smoke plume above northwest France on 29 August was also sampled by LILAS lidar (Hu et al., 2019) at ATOLL observatory in northern France (violet circle in Fig. 6C) several hours after its sampling by the IAGOS flight from Frankfurt. The TROPOMI image taken at 13 UTC (Fig. 6C) shows the plume at the very time of its approach to the LILAS lidar position from northwest.

**Figure 6.** Satellite and airborne observations of Canadian smoke plumes. (A) TROPOMI Absorbing Aerosol Index (AAI) on 11 May 2023, showing smoke plume from the 5 May cluster PyroCb event in Alberta, and IAGOS flight segment A to B color-coded by CO mixing ratio. (B) Time series of flight altitude (color coded by potential vorticity), CO and O<sub>3</sub> mixing ratio measured during the A to B IAGOS flight segment. (C) As in (A) but for 29 August 2023 (WCB uplift episode from 26 August). LILAS lidar location is indicated as violet circle. Radiosonding station location near Paris is shown as black circle. (D) As in B but for the flight in 29 August. Ozone was not measured in this flight.

Figure 7 displays lidar time curtains of backscatter coefficient (Fig. 7A) and volume depolarization ratio (VDR) (Fig. 7C) acquired during the time of plume's transit over the lidar station. The lidar curtains reveal multiple smoke layers throughout the free troposphere with the primary layer extending between 8 and 12 km i.e., across the dynamical tropopause (10.9 km, Fig. 6C), which is aligned with the first thermal tropopause (Fig. 7B). The cloud-free aerosol optical depth (AOD) at 532 nm, computed from lidar extinction below 12 km, varies between 0.9 and 1.2 during the time period 15:15 - 19:30 UTC. This is in close agreement with the collocated AERONET (AEROsol RObotic NETwork) sun photometer reporting the columnar AOD between 1.1 - 1.3. The small difference between AOD measured by LILAS and sun photometer result from incomplete overlap of the lidar system. Due to this issue, extinction coefficient from LILAS was assumed to be constant below 800 m, which may lead to underestimation of AOD. The lidar-derived AOD of the primary smoke layer itself amounts to approximately 1 at 19:00 UTC. For comparison, the highest AOD value within a smoke layer observed over Europe after the PNE wildfire outbreak in August 2017 amounted to 0.6 (Ansmann et al., 2017).

As can be inferred from the black curve in Fig. 7A, the lidar-derived AOD increases further to 2 after 19:30 UTC, however this can be attributed to nucleation and growth of ice cloud particles, as suggested by the sudden increase of backscatter and depolarization within the smoke plume (Fig. 7A, C). The nucleated cirrus appear as wave-like structures, which points to the gravity waves as a trigger for ice nucleation in the smoke-polluted air as has been argued on the bases of a similar lidar observation of cirrus formation inside a smoke plume (Mamouri et al., 2023).

The nucleation of ice particles inside the upper tropospheric smoke plumes may be facilitated by enhanced humidity of the plumes uplifted from the lower troposphere (regardless of the uplift mechanism). We examined high-resolution meteorological profiles from radiosounding data at 23 UTC at Trappes station near Paris, which was aligned with the LILAS lidar in terms of its position with respect to the smoke plume front (cf. Fig. 6C). Figure 7D reports a strong enhancement in relative humidity over ice (RH<sub>ice</sub>) above the vapour saturation within the 8 - 11 km altitude layer. The extent of the hydrated layer correlates well with that of the VDR profile, suggesting that the lidar and the radiosonde have sampled the plume coherently. It is noteworthy that the upper part of the plume (11 - 12 km) is subsaturated (RH<sub>ice</sub> of 40 - 60%), which does not enable cloud particle nucleation and scavenging of smoke aerosols.

Figure 7. Ground-based observation of an intense smoke plume passing above France. on 29 August 2023 (A) Time-altitude curtain of backscatter coefficient at 532 nm from LILAS lidar in Northern France (location shown in Fig. 6C). Black curve indicates the lidar-derived aerosol optical depth (AOD) within 1 - 12.5 km layer (right-hand axis). Black squares represent AOD columnar measurements by collocated AERONET sun photometer. Grey contour indicates the presence of cloud particles detected using depolarization measurements. (B) Vertical profiles of LILAS backscatter coefficient at 532 nm (bottom axis) accumulated during 18:30 - 19:30 period (cloud-free, grey curve) and 21:00 - 22:00 period (aerosol and cloud particles, black curve). Red curve plotted versus top axis shows temperature profile from a radiosounding near Paris launched at 23:15 (all times are UTC). (C) Time - altitude curtain of volume depolarization at 532 nm from LILAS lidar at the ATOLL observatory. Black curve marks the 2 PVU dynamical tropopause. (D) Vertical profiles of LILAS volume depolarization accumulated over 21:00 - 22:00 period (black curve, bottom axis) and relative humidity with respect to ice (RH<sub>ice</sub>, red curve top axis) from a radiosounding near Paris (location of radiosounding station is shown in Fig. 6C).

#### 3.6 Large-scale impact on the stratosphere

In order to quantify and put in perspective the large-scale stratospheric impact of the 2023 wildfire season, we use stratospheric aerosol extinction profiling by Stratospheric Aerosol and Gas Experiment (SAGE) III and OMPS-LP instruments. Figure 8A shows a seasonally- and zonally-averaged aerosol extinction section from SAGE III solar occultation profiles above the local tropopause. The 2023 wildfire stratospheric signal emerges vividly throughout the northern mid- and high latitudes, however its vertical extent is largely limited to the extratropical tropopause layer (ExTL), between 7 - 12 km. The SAGE III latitude-altitude pattern with an enhancement in the ExTL is corroborated by OMPS-LP extinction data (Fig 8C), however the OMPS-LP wildfire perturbation magnitude is a factor of 2.2 smaller compared to that of SAGE III. This is most likely due to the NASA OMPS-LP retrieval assumptions regarding the aerosol microphysical parameters and a related altitude-dependent bias (Chen et al., 2020).

A 7-yr perspective of the 2023 wildfire perturbation is provided in Figs. 8B and 8D, showing height-resolved time series of SAGE III and OMPS-LP aerosol extinction at the northern midlatitudes. The 2023 wildfire signal in the ExTL stands out for its duration and magnitude, being comparable to the perturbation by the Raikoke volcanic eruption in June 2019 (Khaykin et al., 2022; Kloss et al., 2021). However, unlike the Canadian 2017 PNE wildfire outbreak that produced a persistent, confined and self-lofting smoke bubble rising up to 23 km altitude(Lestrelin et al., 2021), the 2023 wildfire-induced stratospheric perturbation is shallow and restricted to altitudes below 12 - 13 km. As can be inferred from Fig. 8B and 8D, smoke pollution of the Northern Summer lower stratosphere is a recurring feature with a variable vertical extent of stratospheric perturbation up to 21 km for PNE outbreak in 2017 and lower for other wildfire seasons. The 2023 stratospheric perturbation, although restricted to the ExTL, is the largest in magnitude and in seasonal extent, spanning 6 months i.e., May through October 2023.

**Figure 8.** Large-scale impact of 2023 wildfires on the stratospheric aerosol load. (A) Latitude altitude section of zonal-mean aerosol extinction at 869 nm from SAGE III averaged over May-October 2023. Black curves mark the mean, minimum and maximum thermal tropopause as well as 380 K isentropic level (see legend). (B) Time-altitude variation of bi-weekly average aerosol extinction over 40° N - 70° N latitude band. Black curves mark the tropopause level (as in A). (C) As in (A) but from OMPS-LP observations. (D) As in (B) but from OMPS-LP weekly averages. Only the stratospheric parts of extinction profiles (above the local thermal tropopause) are shown in all panels.

In order to estimate the mass of smoke aerosols uplifted into the stratosphere during the 2023 wildfire season we use the mass difference method (Khaykin et al., 2020) applied to the global OMPS-LP extinction profiling (see Methods and Fig. S10). Out of the 7 events listed in Table 1, only the three largest ones that occurred during the sequence of WCB uplift episodes (#3, #5 and #6) allowed for a robust estimate of stratospheric aerosol mass perturbation. Their cumulative impact is estimated at 30 - 60 Gg of wildfire aerosol uplifted across the tropopause, which is a lower bound estimate considering the limitations of the mass difference method (see Sect. 2.7). Taking into account the factor of 2.2 underestimation of the ExTL AOD by OMPS-LP as compared to that of SAGE III, the injected masses scale to 0.07 - 0.13 Tg, which is comparable to the largest documented wildfire-induced perturbations, namely the 2009 Black Saturday event (0.05 - 0.1 Tg); the 2017 PNE event (0.1 - 0.3 Tg) and the 2019 ANYSO Phase 1 event (0.2 - 0.8 Tg) as estimated by Peterson et al. (2018; 2020).

## 3.7 Discussion and summary









The 2023 Canadian wildfires have by far exceeded the previous record-breaking events, including the Australian "Black Summer" in terms of the emitted energy (200 TW h) and pyroCb count with a total number of 142 Canadian pyroCb events over the season. The incessant fire activity all across Canada produced a succession of smoke injections into the lower stratosphere.

The pyroCb activity was exceptionally high during the May-July period with an average frequency of 1.4 d<sup>-1</sup>. Nevertheless, only the first cluster pyroCb event in Alberta #1 (5 May) has caused measurable stratospheric smoke pollution, whereas the impact of the other pyroCb events in Canada was limited to the middle and upper troposphere. Peterson et al. (2025) establish that most pyroCbs do not inject material directly into the lowermost stratosphere. Evidence from the 2023 wildfire season supports this, showing that despite the large number of pyroCbs, only a few produced measurable stratospheric injections. Conversely, Peterson et al. (2025) also demonstrate that a season with relatively few pyroCbs can still yield a major injection, as in the 2017 PNE case. The critical factor is the coincidence of favourable meteorological conditions with a sufficiently intense heat flux from the fires. The precise combination of atmospheric dynamics and fire characteristics that enable such major pyroCb-driven stratospheric injections remains poorly understood, underscoring the need for targeted field measurements.

While the pyroCb activity decreased substantially by mid-August, several episodes of significant injections of smoke into the lowermost stratosphere, mostly unrelated to pyroCb activity, could be identified. Using MOCAGE CTM simulations, we showed that the non-pyroCb uplift of smoke from the lower troposphere to the tropopause and above in 2023 owed to the warm conveyor belt (WCB) process. The simulated evolution of the smoke plumes in horizontal and vertical dimensions, as well as its timescale is confirmed by observational data. In contrast to the fast convective uplift by pyroCb events, the WCB process requires 2 - 4 days for the smoke-laden air masses to rise to the tropopause level.

The vertical pathway and its timescale determine the properties and further evolution of stratospheric plumes. The pyroCb development occurs on a scale of a few hours and can drive a volcano-like injection of smoke-icy cloud at the tropopause level (Peterson et al., 2018). With the WCB, the aerosols enter the stratosphere already well mixed and diluted. The low concentration of aerosols in the WCB plumes transported across the tropopause limits the degree of internal heating and thereby does not favor their diabatic self-lofting in the stratosphere, typical for intense pyroCb plumes dynamically confined through the persistent stratospheric anticyclones (SCV or SWIRL). An interesting exception to the SCV self-lofting paradigm is the 5 May pyroCb event in Alberta that produced a compact smoke plume persisting for more than 3 weeks and exhibiting various indications for anticyclonic confinement, but without any signs of self-lofting. A possible reason for the absence of diabatic rise is the relatively low aerosol concentrations in the plume and hence the lack of internal heating. Our results are consistent with cross-tropopause smoke transport in WCBs being predominantly meteorologically driven, while diabatic self-lofting

likely plays only a secondary role under the relatively low aerosol concentrations observed in the upper troposphere. Differences in smoke radiative properties may influence lofting efficiency; however, radiative transfer simulations (Ohneiser et al., 2023) suggest that the absolute concentration of absorbing aerosols is the primary factor. That said, the role of radiatively-driven diabatic self-lofting of smoke in the troposphere requires further investigation.

The lack of diabatic plume rise of the plumes uplifted by WCB events constrained the impact of wildfire emissions to the so-called Extratropical Tropopause Layer (ExTL). The succession of WCB episodes, some of them accompanied by pyroCb activity during the second part of the season, resulted in nearly complete zonal spread of smoke throughout the ExTL north of 40° N in late September- early October.

The bulk of 2023 wildfire smoke pollution was bounded within 9 - 12 km layer, that is at commercial aircraft cruising altitudes. Indeed, the percentage of IAGOS transatlantic flights affected by enhanced CO concentration was a factor of 3 higher than the 20-yr average percentage in the IAGOS CO record since 2003. Some of the IAGOS transatlantic flights sampled extreme CO concentrations exceeding the background levels by a factor of seven. Ground-based lidar measurements in Northern France captured the transit of a dense smoke plume with an extreme AOD of ~1. Such dense smoke layers may present a hazard to commercial aircraft by clogging air filters and coating engine components (Scarbrough, 2023; Veillette, 2021). In addition, flying through thick smoke can affect air quality inside the cabin, posing potential health risks to passengers and crew (Gleim, 2023).

In summary, the extreme 2023 Canadian wildfire season was very different from the previous record-breaking wildfire and pyroCb outbreaks such as PNE and ANYSO that produced long-lived SCVs that self-lofted to the middle stratosphere. PyroCb activity linked to the 2023 wildfires did not produce these self-lofting smoke plumes. However, the incessant fire activity May through September with a succession of WCB episodes during August – September period led to a massive amount of smoke pollution across the Northern Hemisphere extratropical tropopause layer. Smoke aerosols injected at these altitudes can have both direct and indirect radiative effects, which must be examined in future studies to determine the potential impacts on regional and hemispheric radiative balance and weather.

# Acknowledgements








The work was supported by the French ANR (Agence Nationale de la Recherche) PyroStrat 21-CE01- 335 0007-01 project https://pyrostrat.projet.latmos.ipsl.fr/. QH and PG acknowledge support by the OBS4CLIM (Observation for CLIMate) and CaPPA (*Chemical and Physical Properties of the* Atmosphere ) projects supported by the French National Research Agency (ANR) under the France 2030 program, with the reference "ANR-21-ESRE-0013 and "ANR-11-LABX-0005-01" as well as by the Regional Council « Hauts-de-France » and the « European Funds for Regional Economic Development » (FEDER).DP and MF acknowledge the Naval Research Laboratory Base Program for its support of this effort (N0001424WX00026; Dr. J. Hansen). Additional support for D.A.P. was provided by NASA's Modeling, Analysis, and Prediction (MAP) Program (80HQTR21T0099), SAGE III/ISS Science Team (80NSSC24K1183), FireSense Program, and INjected Smoke and PYRocumulonimbus Experiment (INSPYRE).

## **Funding:**

Agence Nationale de la Recherche (ANR) PyroStrat 21-CE01- 335 0007-01 project (SK, SB, SGB, PG, QH, BJ, MM, SP)

CNRS INSU CPJ STANDARDS project (SK)

Agence Nationale de la Recherche ANR-21-ESRE-0013 and ANR-11-LABX-0005-01 (QH, PG)

CNES EXTRA-SAT project (SK, SB, SGB)

Naval Research Laboratory Base Program N0001424WX00026 (DP, MF)

NASA's Modeling, Analysis, and Prediction (MAP) Program 80HQTR21T0099 (DP, MF)

770

780

## **Author contributions**

SK conceived the study and wrote the manuscript. MF, DP, SB, and SGB were involved in the discussion of the results and their interpretation. DP and MF curated and provided the pyroCb-detection data. MM, BJ and SP performed MOCAGE simulations. QH and PG provided LILAS lidar data. AL performed estimation of the injected masses. VT provided IAGOS data. All authors contributed to the final manuscript.

**Competing interests**: The authors declare that they have no competing interests.

## Data and materials availability.

- OMPS-LP data is available at https://snpp
  - omps.gesdisc.eosdis.nasa.gov/data/SNPP\_OMPS\_Level2/OMPS\_NPP\_LP\_L2\_AER\_DAILY.2/2023/; OMPS-NM data are available at https://snpp-
  - omps.gesdisc.eosdis.nasa.gov/data//SNPP\_OMPS\_Level3/OMPS\_NPP\_NMTO3\_L3\_DAILY.2/2023/; SAGE III data at https://doi.org/10.5067/ISS/SAGEIII/SOLAR\_BINARY\_L2-V5.2; TROPOMI data are available at
- <u>https://doi.org/10.5270/S5P-0wafvaf</u>; Meteorological radiosounding data are available at
  - https://donneespubliques.meteofrance.fr/?fond=produit&id\_produit=97&id\_rubrique=33; OHP lidar data are available at https://ndacc.larc.nasa.gov/; LILAS lidar data are available at https://www.icare.univ-lille.fr/asd-
  - $\underline{content/archive? dir=GROUND-BASED/LOA\_Lille/LIDAR-LILAS/GARRLiC\_L2; ERA5\ data\ are\ available\ at$
  - https://doi.org/10.21957/open-data. GFAS data are available at https://ads.atmosphere.copernicus.eu/datasets/cams-global-
- fire-emissions-gfas?tab=download; IAGOS data at <a href="https://iagos.aeris-data.fr/">https://iagos.aeris-data.fr/</a>; MOCAGE simulation data are available through PyroStrat data center upon request. The Worldwide PyroCb Inventory data file used in this study is available as a supplementary data file in Peterson et al. (2025), which has been accepted for publication by npj Climate and Atmospheric Science. Requests for these pyroCb data prior to release by npj Climate and Atmospheric Science should be submitted to: Dr. David Peterson at <a href="https://agos.aeris-data.fr/">data.fr/</a>; MOCAGE simulation data are available through PyroStrat data center upon request. The Worldwide PyroCb Inventory data file used in this study is available as a supplementary data file in Peterson et al. (2025), which has been accepted for publication by npj Climate and Atmospheric Science should be submitted to: Dr. David Peterson at <a href="https://agos.aeris-data.fr/">data.fr/</a>; MOCAGE simulation data are available through PyroStrat data center upon request. The Worldwide PyroCb Inventory data file used in this study is available as a supplementary data file in Peterson et al. (2025), which has been accepted for publication by npj Climate and Atmospheric Science should be submitted to: Dr. David Peterson at <a href="https://agos.aeris-data.fr/">data.fr/</a>; MOCAGE simulation data are available through PyroStrat data. The worldwide PyroCb Inventory data file used in this study is available as a supplementary data file in Peterson et al. (2025), which has been accepted for publication by npj Climate and Atmospheric Science should be submitted to: Dr. David Peterson at <a href="https://aeris.fr/">data.fr/</a> at <a href="https://aeris.fr/">https://aeris.fr/</a> at

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
