# Peer review of "Stratospheric impact of the anomalous 2023 Canadian wildfires: the two vertical pathways of smoke"

_EGUsphere, 2025_

## Referee Comment (RC2)

**Review of "Stratospheric impact of the anomalous 2023 Canadian wildfires: the two vertical pathways of smoke" by Khaykin et al.**

**Reviewer Summary**

This manuscript presents a comprehensive and well-structured analysis of the 2023 Canadian wildfires and their impact on the stratosphere, with a clear distinction between pyrocumulonimbus (PyroCb) and warm conveyor belt (WCB) transport pathways. The study addresses an important topic with high relevance to atmospheric chemistry and climate, particularly in the context of increasing wildfire activity under climate change. The main strength of the paper lies in its careful multi-source observational analysis and event-by-event classification. The primary weakness is the lack of discussion on why the period of peak fire activity did not yield significant stratospheric intrusions, which leaves a gap in the mechanistic understanding. I recommend **acceptance after minor revisions**, provided the authors address the points detailed below.

**General Comments**

The manuscript is well written, logically structured, and addresses an important topic that will likely gain further relevance under ongoing climate change and the projected increase in wildfire frequency and intensity. The analysis of the 2023 Canadian wildfires and their impact on the stratosphere is both thorough and insightful, particularly in distinguishing between PyroCb and WCB pathways. My recommendation is to accept the manuscript for publication in *Atmospheric Chemistry and Physics* after the authors address the following points, which aim to improve the clarity, completeness, and scientific context of the work.

1. **Reconsideration of the PyroCb dominance statement (lines 45–50)**
   In light of the results presented, showing that out of 142 detected PyroCb events, only three (event #1 and partially events #6 and #7; Table 1) were associated with measurable stratospheric impact, the authors may wish to revisit their statement that PyroCb activity is the *primary* source of combustion products entering the stratosphere. It could be valuable to discuss whether the current findings are consistent with this prevailing view or whether alternative pathways, such as WCB transport, might warrant greater emphasis.

2. **Potential human-induced biases in PyroCb detection (Section 2.2)**
   The manuscript notes the use of an "analyst-in-the-loop" approach for PyroCb identification (line 80) in the global database of 761 events (2013–

2023). The authors could elaborate on any potential human-induced biases in this process. In particular, could such biases have contributed to under-detection of PyroCb events during August–October 2023?

3. **Clarification of the SALD product's origin and validation (Section 2.6)**
It is not entirely clear whether the Stratospheric Aerosol Layer Detection (SALD) product, which is derived from OMPS-LP observations, is a locally developed dataset specifically for this study, or an existing product previously used and validated. If the latter, references should be provided; if the former, additional methodological details and validation steps would strengthen the study's reproducibility and robustness.

4. **Missing explanation for lack of stratospheric intrusion during peak fire activity**
While the manuscript clearly distinguishes between PyroCb- and WCB-driven events, it does not address why the period of most intense wildfire activity (June-July), when nearly all PyroCb events were recorded (Fig. 1B) and most of the fire energy was released (Fig. 1A), failed to produce significant stratospheric intrusions.

Possible explanations might include:
a. Reduced intensity or frequency of WCB activity during June–July.
b. A lower tropopause height during September–October, favoring stratospheric intrusion.

These hypotheses could be evaluated using available datasets, and other explanations may also be relevant. To aid such an analysis, it may be useful to:

- Extend Fig. 4's time frame to include May–July.

- Disaggregate Fig. 8A by month or split into May–July and August–October periods.

Without addressing this question, the interpretation of the results remains incomplete, as highlighted by the summary on lines 664–668, which implicitly raises the question of *why* this pattern occurred.

5. **Comparison of CO observations with model output (Section 3.5)**
The analysis in Section 3.5 could be further strengthened by comparing the measured CO concentrations with predictions from the MOCAGE model (if available). Such a comparison would help assess consistency between observations and simulations and provide additional context for interpreting the results.

6. **Inclusion of injected mass estimates for additional wildfire events (lines 650–658)**
Since the authors have already compared their results to other wildfire events (e.g., in Australia), it would be informative to also include the estimated injected aerosol masses for these events, not only for the PNE event. This would provide a more complete comparative framework for evaluating the 2023 Canadian wildfire injections.

**Specific (Technical) Comments**

1. **Spectral range classification** (lines 103–104 vs. lines 113–114)
In lines 103–104, the 300–380 nm range is described as encompassing both "UV and visible spectral regions," whereas in lines 113–114, the 340–380 nm range is referred to solely as "UV spectral bands." The authors should ensure consistent terminology and spectral classification throughout the manuscript.

2. **Reference formatting** (line 131)
The reference to *Taha et al., 2021* appears with inconsistent font formatting. Please standardize to match the manuscript's reference style.

3. **Clarification of Fig. 2C reference** (Section 2.7, line 143)
It is unclear why Fig. 2C is referenced here and what specific information it contributes to this section. Additionally, the relevance of the "16 km" value mentioned in this context should be explained.

4. **Integration of data sources** (end of Section 2)
After introducing all data sources, it would be beneficial to add a brief methodological statement summarizing how these datasets are integrated in the analysis. This would help readers understand the workflow and interconnections between the various observational and model products used.

5. **Justification for WCB diabatic heating statement** (lines 498–499)
The statement that "the low concentration of aerosols in the WCB plumes limits the degree of internal heating and thereby does not enable diabatic self-lofting in the stratosphere" requires either a quantitative calculation or supporting reference to substantiate the claim.

6. **Potential missing context before line 518**
The paragraph beginning at line 518 appears to reference preceding material that is absent. Phrases such as "another CO enhancement…" and "1.5 hours later" clearly indicate continuity with earlier discussion.

The authors should verify whether relevant preceding text has been inadvertently omitted.

---

## Author Response (AR1)

**Reply to Reviewer #1.**

We thank the Reviewer #1 for the positive review and fair remarks, which have all been carefully implemented in the manuscript.

Line 100: Maybe it would be good to change the order of 2.4 TROPOMI and 2.5 OMPS Limb Profiler in order to have OMPS Nadir Mapper and OMPS Limb Profiler in a row.

**Done**

Line 135: How do you justify the thresholds: "0.01 for SAOD and 8 for ER"

These thresholds were empirically defined as a trade-off between the sensitivity to enhanced aerosol layers and minimization of false detections of aerosol layers. The threshold values roughly correspond to seven standard deviations of the zonal-mean values in the non-perturbed conditions. This is now mentioned in the text.

Line 238: North America or Canada?

**Corrected to "Canada"**

Line 252: Is the threshold of AAI > 15 arbitrarily chosen?

This threshold has been previously defined by Peterson et al. (2018) by combining OMPS-NM AAI data and CALIOP aerosol profiling

Table 1: WCB used as acronym but only defined in line 391

**WCB is now defined in the table caption.**

Fig. 1: Why starting at different years 2003, 2013, 2012 for A, B, C and not in the same year for all?

The pyroCb inventory is available only since 2013 whereas the GFAS data set is available since 2003. We included GFAS data prior to 2013 to demonstrate the exceptional character of the 2023 wildfire season. The seasons of interest (represented by coloured curves) have all occurred after 2013 anyway.

Fig. 4: legend caption: kg m³ there is a minus missing before the "3"

**Fixed.**

Fig. 4 and lines 457-463: The events 3, 4, 5, 6 all bring parts of the smoke into the lower stratosphere. You write that the lofting of smoke is meteorologically driven in the WCB and that the diabatic lofting plays a minor role. I understand that vertical transport of smoke in the WCB towards the tropopause is predominantly meteorologically driven. But how can you be sure that the smoke transport through the barrier of the tropopause would happen if it is only meteorologically driven? How would this barrier be passed even in a WCB?

Isentropic mass exchange across the mid-latitude tropopause (e.g., through tropopause folds) is a well-established process. In this context, the upper-level circulation associated with a WCB is expected to modify the tropopause structure on synoptic scales, thereby facilitating cross-tropopause exchange. The ability of the model to reproduce this transport based solely on meteorological reanalysis supports the physical realism of WCB-driven cross-tropopause transport.

How can you exclude that at this point diabatic lofting might dominate to come across the barrier as diabatic lofting plays a big role in the stratosphere?

The role of diabatic lofting driven by solar heating is difficult to quantify without dedicated radiative transfer simulations constrained by multiple observational datasets. Nevertheless, the ability of the meteorologically-driven transport simulation alone to reproduce the uplift of air masses up to the tropopause level suggests that diabatic self-lofting is likely of secondary importance under the relatively low aerosol concentrations observed in the uppermost WCB filaments.

And could differences in the absorptivity of the smoke compared to the Australian wildfire smoke in 2020 explain differences in diabatic lofting behavior?

Variations in the radiative properties of smoke particles, which depend on the type of burned biomass, could indeed influence diabatic lofting. However, basic considerations, supported by radiative transfer simulations (e.g., Ohneiser et al., 2023), suggest that the dominant factor is the concentration of absorbing aerosols rather than their specific absorptivity.

Please discuss this in your manuscript in more detail.

**The following text has been added into the Discussion and summary section:**

Our results are consistent with cross-tropopause smoke transport in WCBs being predominantly meteorologically driven, while diabatic self-lofting likely plays only a secondary role under the relatively low aerosol concentrations observed. Differences in smoke radiative properties may influence lofting efficiency; however, radiative transfer simulations suggest that the absolute concentration of absorbing aerosols is the primary factor. That said, the role of radiatively-driven diabatic self-lofting of smoke in the upper troposphere requires further investigation.

Another question that just comes to my mind: Did you also find smoke transport towards the UTLS (within the tropopshere) if the smoke was not within a WCB (or pyroCb) before?

While the diabatic self-lofting of highly-concentrated smoke plumes in the troposphere cannot be ruled out, all of the observed stratospheric intrusions identified in this study could be associated to either pyroCb- or WCB-driven uplift.

Line 688-690: Too general. In France? Or at that station? Or for Canadian smoke? How can you be sure it was a new record? At least the Australian 2020 smoke had a higher AOD for single layers in the stratosphere.

The mention of the new record has been removed. The 2025 smoke has surpassed this record anyways.

General: Maybe it would be a good idea to include a schematic figure comparing pyroCb and WCB vertical pathways, showing uplift speed, plume structure, and evolution over time.

We have considered several options for such an illustration (including AI-generated renditions) with varying levels of detail, but we were not able to produce a version that we found suitable. Given the considerable uncertainties that remain regarding both pyroCb and WCB dynamics, we believe that such a schematic would be most valuable once the processes are better understood and can be represented with greater confidence.

General: It is good to see that the model could show the lofting of the aerosol in the WCB. It is good to see that the lidar profile shows an AOT of around 1 with a thick smoke plume in the stratosphere. But do you have any case where you also see the observational evidence that the smoke plume does not originate from a pyroCb but was injected at around 2km height in Canada and was later found at a significantly higher altitude? The manuscript would benefit from it.

As a matter of fact, all the non-pyroCb-driven uplift episodes were associated with the smoke that was initially found in the lower troposphere (as the MOCAGE simulation assumed injection altitude of 2 km for all the FRP anomalies). An example of such uplift from the trajectory point of view is provided in Supplementary Fig. S4.

**Technical comments:**

Line 37: Empty space missing after "precipitation"

Line 39: Bracket opened but not closed

Line 56: Which year of the study Peterson et al. ? ("n.d.")

Line 60: Bracket opened but not closed

Line 62: Commas around "however" missing

Line 62: Brackets around Zhang et al need to be removed

Line 78: Brackets around Fromm et al need to be removed

Line 106: Empty space before "OMPS" missing

Line 109: Point missing at the end of the sentence

Line 134: Bracket opened but not closed

Line 168: remove brackets around Guth et al...

Line 184: remove brackets around El Amraoui et al., 2022; Sič et al., 2015

Line 197: remove brackets around (Nédélec et al., 2015)

Line 198: remove brackets around (Blot et al., 2021)

Line 220: remove brackets around Khaykin et al., 2017

Line 392: lofting air instead of lifting

Line 436: these instead of this

Line 505: remove brackets once around Yu et al.

Line 576: 23 UTC, not 23 h UTC

All above has been taken care of.

**Reply to Reviewer #2**

We thank the Reviewer #2 for the positive review and fair remarks, which have all been carefully implemented in the manuscript.

**General Comments**

1. Reconsideration of the PyroCb dominance statement (lines 45–50)
In light of the results presented, showing that out of 142 detected PyroCb events, only three (event #1 and partially events #6 and #7; Table 1) were associated with measurable stratospheric impact, the authors may wish to revisit their statement that PyroCb activity is the *primary* source of combustion products entering the stratosphere. It could be valuable to discuss whether the current findings are consistent with this prevailing view or whether alternative pathways, such as WCB transport, might warrant greater emphasis.

The statement regarding the dominance of the pyroCb pathway (among the other mechanisms invoked) in terms of the large-scale stratospheric impact of wildfires provided in the introduction reflects the state-of-the-art. The most significant large-scale perturbations of stratospheric aerosol load have indeed been unambiguously associated with direct pyroCb injections and this statement still holds true. In this study, we demonstrated the potential of WCB-driven uplift to deliver significant amount of smoke aerosols to the extratropical tropopause layer. The anomalously warm Boreal Summer 2023 characterized by incessant wildfire activity across Canada in combination with the WCB-favorable synoptic conditions over North America during August-September drove a sequence of WCB-driven uplift episodes that had a cumulative effect on the stratosphere comparable with (yet not exceeding) the largest single-event pyroCb injections. The disproportionally smaller stratospheric impact of the numerous pyroCb events during the 2023 season does not necessarily undermine the dominance of the pyroCb pathway. The physics and lifecycle of extreme pyroconvection are still poorly constrained and many questions remain regarding the factors and conditions permitting major pyroCb injections into the stratosphere.

2. Potential human-induced biases in PyroCb detection (Section 2.2)
The manuscript notes the use of an "analyst-in-the-loop" approach for PyroCb identification (line 80) in the global database of 761 events (2013–2023). The authors could elaborate on any potential human-induced biases in this process. In particular, could such biases have contributed to under-detection of PyroCb events during August–October 2023?

There is a chance for missing a pyroCb by the analyst-in-the-loop process as a general matter. This even occurred in 2023: one additional pyroCb, in September, was identified months after the 2023 season end. However, for the period in question, that being mid-July to late-August 2023, we claim no bias potential because our analysis of

the curiously strong lower stratospheric aerosol signals reported in the manuscript involved multiple revisits of GOES imagery wherever fire hot-spots were in evidence for the unmistakable microphysical signals of a pyroCb. It was a review like this that led to our post-season detection of the single additional September pyroCb mentioned above. There were no relevant gaps in GOES coverage that could have made our search vulnerable to a missed pyroCb detection. We recognize the imperfect nature of analyst-in-the-loop pyroCb detection for domains and times for which no such scrutiny is demanded. But in this case, we have high confidence that minimal bias due to false negative or false positive detection occurred. The details and caveats of pyroCb detection are provided by Peterson et al. (2025), npj Climate and Atmos. Sci. (accepted).

3. Clarification of the SALD product's origin and validation (Section 2.6) It is not entirely clear whether the Stratospheric Aerosol Layer Detection (SALD) product, which is derived from OMPS-LP observations, is a locally developed dataset specifically for this study, or an existing product previously used and validated. If the latter, references should be provided; if the former, additional methodological details and validation steps would strengthen the study's reproducibility and robustness.

The 'enhanced aerosol layer' flag, supplied with the NASA OMPS-LP retrieval, is described by Taha et al. (2021). The SALD product is the same but with additional filtering applied to minimize false detections of stratospheric aerosol layers linked to e.g., high-altitude clouds and small-scale tropopause altitude variations. We added some details regarding SALD definition to the respective section (Sect. 2.6). The reproducibility of this data product should be straightforward.

4. Missing explanation for lack of stratospheric intrusion during peak fire activity

While the manuscript clearly distinguishes between PyroCb- and WCB- driven events, it does not address why the period of most intense wildfire activity (June-July), when nearly all PyroCb events were recorded (Fig. 1B) and most of the fire energy was released (Fig. 1A), failed to produce significant stratospheric intrusions.

There is an emerging realization of the fact that there is limited correlation between the basic wildfire metrics (such as burned area, FRP or pyroCb count) – and the magnitude of stratospheric perturbation.

It is established by Peterson et al. (2025) that the preponderance of pyroCbs do not inject material explicitly into the lowermost stratosphere. It is reasonable to state that the evidence of the 2023 wildfire season is that despite the large number of pyroCbs, the number of measurable stratospheric injections was small. Conversely, Peterson et al. (2025) also show that one does not need a season with many pyroCbs to have a major stratospheric injection (e.g., 2017 PNE in Canada). The key to a significant stratospheric impact the right set of meteorological conditions to be present

at the same time as a significant heat flux from the fire. The fine details of the meteorological conditions and fire characteristics involved in major stratospheric injections is yet to be understood and this is where field measurements are required. A comprehensive field experiment dedicated to pyroCb is planned in 2026 within the NASA INSPYRE project.

Possible explanations might include:

a. Reduced intensity or frequency of WCB activity during June

–July.

Climatologically, the WCB activity is lower during early Summer (Eckhardt et al., 2004)

b. A lower tropopause height during September–October, favoring stratospheric intrusion.

While the seasonal variation of tropopause altitude may be deemed an important factor from the general point of view, the seasonality of the most significant stratospheric perturbations over the last decade does not support this conjecture suggesting a complexity of the processes involved.

These hypotheses could be evaluated using available datasets, and other explanations may also be relevant. To aid such an analysis, it may be useful to:

- Extend Fig. 4's time frame to include May–July.
   Unfortunately, the simulation output obtained using the current model configuration is only available for August October 2023 period.
  - Disaggregate Fig. 8A by month or split into May–July and August– October periods.

We believe that the altitude reach of aerosol perturbation month-by-month is readily inferable from Fig. 8b,d.

Without addressing this question, the interpretation of the results remains incomplete, as highlighted by the summary on lines 664–668, which implicitly raises the question of *why* this pattern occurred.

The following text was added into the respective paragraph in Sect. 3.7:

Peterson et al. (2025) establish that most pyroCbs do not inject material directly into the lowermost stratosphere. Evidence from the 2023 wildfire season supports this, showing that despite the large number of pyroCbs, only a few produced measurable stratospheric injections. Conversely, Peterson et al. (2025) also demonstrate that a season with relatively few pyroCbs can still yield a major injection, as in the 2017 PNE case. The critical factor is the coincidence of favourable meteorological conditions with a sufficiently intense heat flux from the fires. The precise combination of atmospheric dynamics and fire characteristics that enable such major pyeoCb-driven stratospheric injections remains poorly understood, underscoring the need for targeted field measurements.

The text in the last paragraph was modified as follows: "In summary, the extreme 2023 Canadian wildfire season was very different from the previous record-breaking wildfire and pyroCb outbreaks such as PNE and ANYSO that produced long-lived SCVs

that self-lofted to the middle stratosphere. PyroCb activity linked to the 2023 wildfires did not produce these self-lofting smoke plumes. However, the incessant fire activity May through September with a succession of WCB episodes during August – September period led to a massive amount of smoke pollution across the Northern Hemisphere extratropical tropopause layer."

**5. Comparison of CO observations with model output (Section 3.5)**

The analysis in Section 3.5 could be further strengthened by comparing the measured CO concentrations with predictions from the MOCAGE model (if available). Such a comparison would help assess consistency between observations and simulations and provide additional context for interpreting the results.

The simulated CO data are not available from this particular modeling experiment. The comparison between MOCAGE simulations and IAGOS CO observations is provided by Cussac et al. (2020). As far as the consistency between MOCAGE and observations (in terms of the plume pattern and altitude) involved in this study, this is demonstrated in Fig. 3 and Fig. 4. Another study in progress by Hu et al. will provide a detailed comparison of MOCAGE simulations and lidar observations.

**6. Inclusion of injected mass estimates for additional wildfire events (lines 650–658)**

Since the authors have already compared their results to other wildfire events (e.g., in Australia), it would be informative to also include the estimated injected aerosol masses for these events, not only for the PNE event. This would provide a more complete comparative framework for evaluating the 2023 Canadian wildfire injections.

**The text has been revised as follows:**

Taking into account the factor of 2.2 underestimation of the ExTL AOD by OMPS-LP as compared to that of SAGE III, the injected masses scale to 0.07-0.13 Tg, which is comparable to the largest documented wildfire-induced perturbations, namely the 2009 Black Saturday event (0.05-0.1 Tg); the 2017 PNE event (0.1-0.3 Tg) and the 2019 ANYSO Phase 1 event (0.2-0.8 Tg) as estimated by Peterson et al. (2018; 2020).

**Specific (Technical) Comments**

Spectral range classification (lines 103–104 vs. lines 113–114)
 In lines 103–104, the 300–380 nm range is described as encompassing both "UV and visible spectral regions," whereas in lines 113–114, the 340–380 nm range is referred to solely as "UV spectral bands." The authors should ensure consistent terminology and spectral classification throughout the manuscript.

**The mention of visible spectral range is incorrect and has been removed.**

**2. Reference formatting (line 131)**

The reference to *Taha et al., 2021* appears with inconsistent font formatting. Please standardize to match the manuscript's reference style.

Formatting fixed.

**3. Clarification of Fig. 2C reference (Section 2.7, line 143)**

It is unclear why Fig. 2C is referenced here and what specific information it contributes to this section. Additionally, the relevance of the "16 km" value mentioned in this context should be explained.

The reference was corrected (Fig. S10).

**4. Integration of data sources (end of Section 2)**

After introducing all data sources, it would be beneficial to add a brief methodological statement summarizing how these datasets are integrated in the analysis. This would help readers understand the workflow and interconnections between the various observational and model products used.

**A subsection has been added:**

**2.13 Integration of data sources**

In this study, pyroCb detections from the global inventory were combined with satellite observations from OMPS-NM, TROPOMI, and OMPS-LP to track smoke injection and transport. The stratospheric extinction profiles from OMPS-LP and SAGE III/ISS were used to constrain the large-scale aerosol perturbation. Ground-based lidar (LILAS, OHP) and radiosonde profiles provided high-resolution vertical structure, while IAGOS in situ aircraft data supplied CO and O3 measurements for characterization of plume chemical composition. These observational datasets were combined with fire emissions from GFAS and compared against MOCAGE chemistry-transport simulations (with MODIS AOD assimilation) to evaluate injection heights, aerosol loading, and plume dispersion. This integrated workflow provides a consistent observational—model framework for analyzing the evolution of wildfire smoke in the lower stratosphere.

**5. **Justification for WCB diabatic heating statement** (lines 498–499)**

The statement that "the low concentration of aerosols in the WCB plumes limits the degree of internal heating and thereby does not enable diabatic self-lofting in the stratosphere" requires either a quantitative calculation or supporting reference to substantiate the claim.

The following sentence has been added here: Indeed, radiative transfer simulations by Ohneiser et al. (2023) showed that the lofting rate strongly depends on the smoke plume's AOD.

**6. Potential missing context before line 518**

The paragraph beginning at line 518 appears to reference preceding material that is absent. Phrases such as "another CO enhancement..." and "1.5 hours later" clearly indicate continuity with earlier discussion. The authors should verify whether relevant preceding text has been inadvertently omitted.

**There is indeed a large chunk of text missing, it has been recovered:**

During the active wildfire season in 2023, May through September, the IAGOS flights covered a total travel distance of 4.3 million km at cruising altitudes within the outflow region of Canadian wildfires  $(40^{\circ}N - 90^{\circ}N, 130^{\circ}W - 30^{\circ}E)$ , out of which 8244 km (0.19%), that is ~34 hours of flight time, was spent in conditions with CO concentration exceeding +3 sigma limit (195 ppbv, computed from the ensemble of cruise data). The percentage of transatlantic IAGOS flights affected by enhanced CO concentration amounts to 0.19%, which is a factor of 3 higher than the 21-yr average percentage of 0.06% (Fig. S9).

Figure 6 shows two examples of transatlantic flights sampling intense smoke plumes from high-resolution Sentinel 5P TROPOMI AAI observations. The first case of 11 May 2023 corresponds to a flight from Montreal that crossed a 6-day old plume originating from the cluster pyroCb event in Alberta on 5 May (#1 in Table 1). The flight track across the high-AAI plume over Nova Scotia is shown in Fig. 6A, whereas the time series of GPS altitude, CO and O3 mixing ratio along the A-B flight segment are shown in Fig. 6B. Shortly after reaching the cruise altitudes and crossing the dynamical tropopause (2 PVU), the aircraft was exposed to high CO mixing ratios reaching 601±39 ppbv. The CO enhancements are correlated with substantial dips in ozone mixing ratio, depleted by a factor of 4 with respect to the extra-plume environment. The ozone depletion within the smoke plumes has been reported by Bernath et al. (2022); Solomon et al. (2023); Ohneiser et al. (2021) and can be associated with transport and/or chemical processes.

**Reply to Reviewer #3**

We thank the Reviewer #3 for the positive review and fair remarks, which have all been taken care of in the revision.

General comments:

Section 2.2: I would suggest to add in the supplementary material some illustration of a PyroCb as identified from brightness temperature.

Such an example along with a detailed description of the pyroCb detection method is provided by Peterson et al., 2025, which has been accepted for publication in npj Climate and Atmos. Sci. journal.

Section 2.5: Please provide more details about OMPS-LP NASA v2 uncertainty which depends on the wavelength as explained in Taha et al. (2021). The choice of the 869 nm channel can be explained.

**Added a sentence:** "The 869 nm channel is chosen because it showed the best agreement with SAGE III data (Taha et al., 2021)."

Line 150: I am surprised that the Shiveluch volcanic eruption in 2023 was significant enough to impact the stratospheric aerosol content on a seasonal scale. From OMPS-LP observations, my view is that this signal was unclear at the hemispheric scale and drowned by the (slight) propagation of the Hunga plume to the NH extratropical latitudes. I would suggest to tone down the statement about the impact of this eruption.

Indeed, the effect of Shiveluch eruption is small on a seasonal scale. The mention of this event has been removed.

Copernicus Atmosphere Monitoring Service (2020): CAMS global emission inventories. Copernicus Atmosphere Monitoring Service (CAMS) Atmosphere Data Store, DOI: 10.24381/1d158bec (Accessed on 12-Sep-2025).

Section 3.1 and figure 1: Fire cumulative energy is increasing steadily throughout the season while pyroCb cumulative counts show different evolution with for instance stable values (around 130) in August 2023. What is the reason for such difference? Is it because fire cumulative energy also accounts for fires without pyroCb formation?

The fire cumulative energy, which is proportional to the burned area, is not expected to correlate closely with the pyroCb count. The occurrence of pyroCb strongly depends on the meteorological conditions as pointed out by Peterson et al. (2025). The physics and lifecycle of extreme pyroconvection are still poorly constrained and many questions remain regarding the factors and conditions permitting major pyroCb injections into the stratosphere.

Section 3.2: I do not clearly grasp how the authors find pyroCb cases organizing as clusters. I guess this is done from geostationary imaging but the reader does not have any illustration or explanation. Does it correspond to pyroCb occurring at the same time nearby or at far distances?

The cluster events are defined as 3 or more individual pyroCbs occurring within a 3° x 3° deg. domain and 24 hours (Sect. 3.2.).

Also, how do the authors estimate a "measurable stratospheric impact" from 7 specific events? It is not obvious to derive a number of 7 events reaching at least ~10 km in altitude in Figure 2A and 2B since, as stated in the manuscript, a given plume can survive transport more than twice around the globe and other plumes, freshly emitted, can complexify the readability of Figure 2. To clarify these points, perhaps an additional figure with brightness temperature (BT) could be helpful.

Indeed, it is not obvious to derive all the 7 events listed in Tab. 1 from Fig. 2. This is why we provide further support for the attribution of stratospheric plumes to specific wildfires as a sequence of daily AAI maps with SALD and pyroCb locations in Supplementary Animation 1. The BT would only pick pyroCb events and not necessarily reaching the stratosphere. In this respect AAI and SALD represent a much better metrics.

Section 3.3: this is not a major issue but from Figure 3, the model seems to mostly underestimate the altitude of the aerosol layer peak. This is a bit surprising since the timing of transport and the overall horizontal distribution of the plume is rather well reproduced by the model. Some improvement can be seen approaching 17/08. Is this due to inadequate injection height (constrained to 2 km everywhere) and/or issues regarding vertical transport of particles (e.g. resulting from WCB representation in ARPEGE). Did the authors test higher injection heights with some information taken from BT? Another possibility could be radiative lofting process of optically-absorbing smoke aerosols which is not computed in the model, although the uplift rate through this mechanism is expected to be lower.

The apparent underestimation of the aerosol peak by MOCAGE may be simply due to methodological issue. The SALDs are limited to OMPS-LP profiles featuring an enhancement above the tropopause, whereas the MOCAGE-derived aerosol peak altitude may be low-biased due to tropospheric plumes which are filtered out in the OMPS-LP retrieval. We did test various injection heights for the simulation however the effect on the plume top altitude was very small.

Lines 393-397: Some of the information about MOCAGE given here is redundant with Section 2.9. Please simplify accordingly.

**Done.**

In section 3.4, the authors propose that differences in aerosol concentrations can be the reason for differences in plume altitudes and time evolution for Pyrocb versus WCB. Could we have any comparison with large reported events (ANYSO and PNE events) in term of ER to emphasize the lower amplitude of the 2023 episodes? Also, the idea of the authors is plausible but no mention of any role of aerosol microphysics is provided. Bigger particles would give higher ER but would not match the lower ER for WCB. In this case, coalescence could increase the sizes of the particles but would reduce concentrations. Could MOCAGE provide some indications about aerosol microphysical evolution (I see there are 6 bins in the microphysical module)? We have a few information in the literature on smoke particle sizes from Pyrocbs. From the FIREX-AQ data (see Peterson et al. BAMS, 2022), aerosol and cloud particle sizes within a pyroCb are not static which could explain the differences the Alberta and Siberia cases. They evolve vertically due to microphysical processes (freezing, coalescence) and temporally as the plume ages. The efficiency of the pyroCb as a smoke injection mechanism could be intrinsically linked to these particle sizes through the precipitation scavenging feedback loop. Larger particles promote scavenging and less efficient transport, while smaller particles, fostered by intense updrafts, facilitate massive injection of smoke into the stratosphere. I suggest these elements to be clarified (or ruled out) in the text in a few sentences.

Thank you for this insight. The microphysical properties and their evolution in pyroCb plumes may indeed play a role. However, basic considerations, supported by radiative transfer simulations (e.g., Ohneiser et al., 2021), suggest that the dominant factor is the concentration of absorbing aerosols rather than their size distribution or specific absorptivity. Without information on the particle composition of Alberta and Magadan plumes we cannot factor in the microphysics as a possible reason for the difference in behaviour. However, what can be inferred directly from the available observations is that the Magadan plume featured higher peak ER values.

**We added a sentence in the end of Sect. 3.4:** "For comparison, the peak ER values of the **voung** PNE plumes reached 74."

Line 518: please provide the date of the case discussed here.

There was a chunk of text missing in this section, this was fixed.

Technical errors:

Line 33: correct to Salawitch and McBride, 2022

Line 114, 116: Choose the acronym AAI instead of AI for homogeneity throughout the manuscript.

Line 56 (and in other locations in the manuscript): Peterson et al., n.d. Do you have the year of this publication? Is it in open access? I cannot find the reference.

Line 126: SALD acronym is supposed to be for Stratospheric Aerosol Layer Detection and is defined in the title of section 2.6 which is enough to me.

Line 137: add the bracket after "maximum ER"

Line 149: Add "stratosphere" after global.

Lines 174-175: correct to Bechtold et al. (2001) and Louis (1979).

Line 203: add "s" to "wavelength".

Section 2.12 title: define acronym for LTA

Figure 2: use the term AAI instead of AI in the figure axis title of Figure 2C. Within the figure, individual pyroCb events are not marked as a small but as an open triangle conversely to what is indicated in the caption.

Line 423: correct to "analyzed"

Line 518: the date of the first event from Figure 6A is missing in the text.

Line 521: correct to "a SCV-like"

All the above have been fixed.

**Reply to Reviewer #4**

We thank the Reviewer #4 for the positive review and insightful remarks.

**General Comments**

The authors argue that during the Canada wildfire season in 2023, uplifting of smoke into the lower stratosphere was mainly caused by a WCB mechanism rather than by diabatic heating (and self-lofting) of air masses laden with smoke particles. However, is pyroCb activity still required for the WCB mechanism to effectively raise particles into the lower stratosphere?

According to our findings, the PyroCb activity is not required for the WCB to effectively rise lower-tropospheric smoke to UTLS. The pyroCb uplift is primarily driven by deep-convective process augmented by the energy of combustion as explained in Sect. 3.7. Otherwise said, while the pyroconvection may generate short-term (yet massive) cross-tropopause overshoots, the WCB acts to gradually lift the warm airmasses into the upper troposphere, which can eventually penetrate into the stratosphere.

In order to clarify the swiftness of the pyroCb process (in contrast to WCB), the respective text has been revision in the Discussion and summary section:

"The pyroCb development occurs on a scale of a few hours and can drive a volcano-like injection of smoke-icy cloud at the tropopause level (Peterson et al., 2018). With the WCB, the aerosols enter the stratosphere already well mixed and diluted."

In the meteorological WCB circulation over the region, what was the lowest altitude at which smoke particles could have entered the conveyor belt? Were smoke plumes not associated with pyroCb activity high enough?

The MOCAGE simulation was initialized with smoke injections at or below 2 km altitude, i.e. without any account for pyroCu or pyroCb events whatsoever as mentioned in Sect. 2.9. The ability of the model to reproduce this transport based solely on meteorological reanalysis supports the physical realism of WCB-driven cross-tropopause transport.

If diabatic heating is not at all necessary in the WCB mechanism, in order to breach through the upper troposphere smoke particles need to be carried by the WCB circulation.

As a matter of fact, the isentropic mass exchange across the mid-latitude tropopause (e.g., through tropopause folds) is a well-established process that does not require external radiative forcing. In this context, the upper-level circulation associated with a WCB is expected to modify the tropopause structure on synoptic scales, thereby facilitating cross-tropopause exchange.

**The following text has been added into the Discussion and summary section:**

Our results are consistent with cross-tropopause smoke transport in WCBs being predominantly meteorologically driven, while diabatic self-lofting likely plays only a secondary role under the relatively low aerosol concentrations observed. Differences in smoke radiative properties may influence lofting efficiency; however, radiative transfer simulations suggest that the absolute concentration of absorbing aerosols is the primary factor. That said, the role of radiatively-driven diabatic self-lofting of smoke in the upper troposphere requires further investigation.

Does the WCB circulation always penetrates the lower stratosphere? If so, some discussion on this point is necessary. If not, at what altitudes were smoke particles delivered and how did they cross into the lower stratosphere?

The vertical extent of a WCB-driven uplift depends on the environmental synoptic conditions and it is conceivable that the preponderance of WCB do not reach the tropopause levels. However, in the cases observed in 2023 demonstrate that such a deep WCB uplift from the lower troposphere (~2 km) is possible.

If the WCB mechanism can efficiently lift smoke particles into the lower stratosphere, it should be able to do the same with other particulate species. Is there any evidence of non-black/brown carbon aerosols carried into the lower stratosphere by this process?

There exists evidence for the WCB-driven uplift of desert dust up the tropopause levels. The phenomenon is termed dust-infused baroclinic storm (DIBS) (Fromm et al., GRL, 2016), which is cited in our paper.

The authors argue that lifting timescales differ between diabatic self-lofting and WCB-driven injection of stratospheric smoke. Are the altitudes of injection also different? Would a stratospheric smoke layer delivered by WCB be lower than one delivered through diabatic self-lofting? If so, could this difference be used to constrain the lifting process?

There might be a misunderstanding. We refer to diabatic self-lofting with regard to highly-concentrated smoke plumes directly injected into the UTLS by pyroCbs. As discussed in Sect. 3.7, "The vertical pathway and its timescale determine the properties and further evolution of stratospheric plumes. With the WCB, the aerosols enter the stratosphere already well mixed and diluted. The low concentration of aerosols in the WCB plumes transported across the tropopause limits the degree of internal heating and thereby does not favor their diabatic self-lofting in the stratosphere, typical for intense pyroCb plumes...".

The altitude of smoke layer can provide a clue on the uplift mechanism however to constrain it, one has to track the smoke plumes in time and space from their source location and dispose of information on the pyroCb events. That being said, a smoke layer several kilometers above the tropopause may be unambiguously linked with a massive pyroCb injection followed by diabatic self-lofting in the stratosphere.

**Minor corrections:**

A number of citations have extra parentheses. For example, on line 20 "(e.g. (Khaykin et al., 2020; Peterson et al., 2021)." Other occurrences appear throughout the text.

On line 300: "...were linked respectfully to..." -> "...were linked respectively to..."?On line 661: "...in terms of the emitted power" should be "...in terms of the emitted energy", since the quantity is a "TW h".

All the above correction applied.